# Mapping QTL for Adult-Plant Resistance to Stripe Rust in a Chinese Wheat Landrace

**DOI:** 10.3390/ijms23179662

**Published:** 2022-08-26

**Authors:** Yunlong Pang, Chunxia Liu, Meng Lin, Fei Ni, Wenhui Li, Jin Cai, Ziliang Zhang, Huaqiang Zhu, Jingxian Liu, Jiajie Wu, Guihua Bai, Shubing Liu

**Affiliations:** 1State Key Laboratory of Crop Biology, College of Agronomy, Shandong Agricultural University, Tai’an 271018, China; 2Department of Agronomy, Kansas State University, Manhattan, KS 66506, USA; 3Institute of Germplasm Resources and Biotechnology, Jiangsu Academy of Agricultural Sciences, Nanjing 210014, China; 4Hard Winter Wheat Genetics Research Unit, Manhattan, KS 66506, USA

**Keywords:** wheat, stripe rust, adult plant resistance, candidate gene, KASP

## Abstract

Wheat stripe (yellow) rust is a worldwide disease that seriously reduces wheat grain yield and quality. Adult-plant resistance (APR) to stripe rust is generally more durable but usually controlled by multiple genes with partial resistance. In this study, a recombinant inbred line population was developed from a cross between a Chinese wheat landrace, Tutoumai, with APR to stripe rust, and a highly susceptible wheat cultivar, Siyang 936. The population was genotyped by genotyping-by-sequencing and phenotyped for APR to stripe rust in four consecutive field experiments. Three QTLs, *QYr.sdau-1BL*, *QYr.sdau-5BL*, and *QYr.sdau-6BL*, were identified for APR to stripe rust, and explained 8.0–21.2%, 10.1–22.7%, and 11.6–18.0% of the phenotypic variation, respectively. *QYr.sdau-1BL* was further mapped to a 21.6 Mb region using KASP markers derived from SNPs identified by RNA-seq of the two parents. In the *QYr.sdau-1BL* region, 13 disease-resistance-related genes were differently expressed between the two parents, and therefore were considered as the putative candidates of *QYr.sdau-1BL*. This study provides favorable gene/QTL and high-throughput markers to breeding programs for marker-assisted selection of the wheat stripe rust APR genes.

## 1. Introduction

Wheat stripe rust or yellow rust (YR), caused by *Puccinia striiformis f.* sp. *tritici* (*Pst*), seriously threatens wheat production worldwide [1]. Stripe rust can result in 10–70% wheat yield losses in susceptible cultivars under prevailing climatic conditions. However, in severe epidemic years, some fields can have yield losses up to 100% in susceptible cultivars due to YR infection [2]. Large economic losses caused by YR epidemics usually occur in the major wheat-producing countries such as Ethiopia, USA, Australia, and China [1]. In China, YR occurs in about 4.1 million hectares each year, which poses a serious threat to Chinese wheat production [3]. 

Application of fungicides can reduce the losses caused by YR; however, developing YR-resistant cultivars is a more economic, effective, and environmentally friendly approach [4,5]. The YR resistance can be categorized into all-stage resistance (ASR) and adult-plant resistance (APR). ASR shows resistance at both the seedling and adult-plant stages, and the resistance is usually race specific and controlled by major genes. However, ASR is more vulnerable to new pathotypes from mutation, recombination, or migration [6,7]. APR often exhibits incomplete resistance at the post-seedling growth stages, is non-race specific or spectrum resistant, and is controlled by multiple quantitative trait loci (QTLs) [8]. 

However, most of the cataloged *Yr* genes (*Yr1*–*Yr83*; *YrU1*) to date are race-specific ASR [9] due to the complex quantitative inheritance of APR. Many cataloged ASR genes are no longer effective or likely to have a short duration of effectiveness if widely deployed in commercial wheat production [10,11]. Thus, APR is drawing more and more attention in breeding programs to improve YR resistance in wheat [5]. 

The recent advancement in genome-wide single nucleotide polymorphism (SNP) genotyping technologies, such as genotype-by-sequencing (GBS) [12] and high-density SNP chips [13,14], have revolutionized the gene cloning or QTL mapping in wheat, which facilitates quick APR-gene identification and marker development for YR resistance. In a recombinant inbred line (RIL) population genotyped by 4662 high-quality GBS-SNP markers, two QTLs for YR APR on chromosomes 3B and 3D were identified [15]. Two QTLs for YR APR, *QYrto.swust-3AS* and *QYrto.swust-3BS*, were identified in a RIL population genotyped by 35K SNP arrays [16]. Six QTLs for YR APR located on 1BL, 2AS, 4BS, 5AL, 6DS, and 7BL were detected in a RIL population genotyped by 55K SNP arrays, and a kompetitive allele-specific PCR (KASP) marker for *QYr.hebau-5AL* was successfully developed [17]. The conversion of SNPs into breeder-friendly KASP markers will greatly underpin the efficiency and cost-effectiveness of marker-assisted selection (MAS) in wheat breeding [18].

In this study, we developed a RIL population from a cross between the APR landrace Tutoumai (TTM) and susceptible cultivar Siyang 936 (SY936), and genotyped the population using GBS-SNPs. The objectives of this study were to (1) map QTLs for YR APR in TTM and develop closely linked KASP markers for breeding; (2) identify putative genes regulating YR APR using RNA-seq; and (3) select elite YR-resistant lines as potential germplasm for wheat breeding.

## 2. Results

### 2.1. Phenotypic Variation and Correlation

At the seedling stage, both SY936 and TTM were highly sensitive to YR; at the adult-plant stage, however, TTM became highly-resistant whereas SY936 remained highly sensitive, indicating APR to YR in TTM (Figure 1). TTM were scored 1 for IT, 1–5 for MDS, and 10–200 for AUDPC; whereas SY936 had an IT of 3–4, MDS of 80–100, and AUDPC of 500–600. The IT, MDS, and AUDPC of the RILs ranged from 0 to 4, 1 to 100%, and 10 to 1725.5, respectively, under different environments (Figure 2). All three traits showed continuous distributions with considerable variation in the population, and transgressive segregation, indicating that the sensitive parent SY936 might also contribute favorable alleles to YR resistance. Due to the severe epidemic of YR in 2019, the mean IT (3.2), MDS (59.7), and AUDPC (400.4) in the 2019 experiment was slightly higher than those from the 2016 (IT = 2.3, MDS = 33.8, AUDPC = 224.6), 2017 (IT = 2.1, MDS = 32.0, AUDPC = 216.7), and 2018 (IT = 2.3, MDS = 28.8, AUDPC = 2051) experiments.

Every YR trait showed moderate to high correlations with each other among the experiments, with r-values ranging from 0.79 between IT and AUDPC in 2019 to 0.97 between MDS and AUDPC in 2016 (Appendix A). ANOVA indicated that genotypes of RILs accounted for most of the phenotypic variations for all the traits, followed by environment and genotype-by-environment interaction (Table 1). The heritability was 0.83, 0.80, and 0.83 for IT, MDS, and AUDPC, respectively.

### 2.2. QTL Mapping

Three QTLs for YR APR were identified on chromosomes 1BL, 5BL, and 6BL (Figure 3 and Table 2). *QYr.sdau-1BL* was detected in the 2016, 2017, and 2018 experiments for all three traits, and explained 8.0–13.8%, 10.8–17.7%, and 9.2–21.2% of the phenotypic variation for IT, MDS, and AUDPC, respectively. *QYr.sdau*-*5BL* was detected in the 2016 and 2018 experiments, with phenotypic variation explained (PVE) of 2.6–4.0%, 2.6–5.9%, and 3.0–4.6% for IT, MDS, and AUDPC, respectively. *QYr.sdau-6BL* was detected in all the four experiments, with a PVE of 16.5–18.0%, 12.1–13.7%, and 11.6–16.4% for IT, MDS, and AUDPC, respectively. Among the three QTLs, the resistance alleles of *QYr.sdau-1BL* and *QYr.sdau-6BL* were from TTM, while *QYr.sdau*-*5BL* was from SY936 (Table 2).

### 2.3. Additive Effect of Identified QTLs

In general, the RILs harboring none of the resistance alleles at the three QTLs were YR susceptible, with the mean IT, MDS, and AUDPC being 3.1–3.7, 42.5–77.4, and 284.9–534.3, respectively, across the four years. For the RILs harboring only one of the three QTLs, the resistance was slightly better than the RILs with none of the resistance alleles, indicating a single QTL provided only partial resistance. For the RILs with two or three QTLs, the level of resistance was considerably higher than those with only one resistance allele, and the mean IT, MDS, and AUDPC for the RILs with all three QTLs were 1.1–2.5, 5.5–36.1, and 48.1–252.7, respectively, across the four years (Figure 4). 

### 2.4. RNA-Seq Analysis of Two Parents

After removal of the low-quality data, adapter reads, and rRNA, a total of 11.2 Gb and 11.4 Gb clean sequence data were obtained for TTM and SY936, containing 74,978,480 and 76,031,092 clean reads, respectively. The clean reads from the two samples accounted for 99.69% of the total raw data with Q30 > 93.56%, and the GC content of 52.81% and 53.42% for TTM and SY936, respectively. By aligning each of the two sets of clean reads to IWGSC Refseq v 1.1, about 93% of the reads were mapped on the reference sequence (Appendix A).

In total, 32,499 SNPs were identified between TTM and SY936, with most of variants (72.7%) in the coding DNA regions (CDS) and untranslated regions (UTR). The number of SNPs on each chromosome ranged from 174 on 4D to 3227 on 5B (Appendix A). Among the 67,676 genes identified in the two parents, 8097 genes were downregulated and 7553 genes were upregulated in SY936 (Figure 5A). GO analysis indicated that these differently expressed genes (DEGs) were mainly involved in cell components, including the cytoplasmic membrane-bounded vesicle, integral component of membrane, and membrane; in molecular functions, including nucleotide binding, protein kinase activity, and the structural constituent of ribosomes; and in biological processes, including the metabolic process and oxidation–reduction process (Figure 5B). The KEGG analysis showed that most of the DEGs were enriched in the pathway of the biosynthesis of secondary metabolites, followed by plant−pathogen interaction, ribosome, and phenylpropanoid biosynthesis (Figure 5C). In particular, 525 upregulated and 360 downregulated genes participating in the plant–pathogen interaction pathways were identified, which suggested key regulatory genes responsible for YR resistance (Figure 5D).

### 2.5. Enhancing Marker Density in QYr.sdau-1BL Linkage Map

*QYr.sdau-1BL* was mapped between marker TTM_253857_21 and TTM_542992_8, spanning a physical interval from 653.0 to 682.1 Mb on chromosome 1BL (Figure 6A). To further narrow down the interval of *QYr.sdau-1BL*, five KASP markers were developed within the region based on the SNPs identified in this interval by RNA-seq (Figure 6B). Using the linkage map re-constructed using the five RNA-seq-derived KASP markers together with the GBS-generated markers, *QYr.sdau-1BL* was further mapped between KASP_63005 at 655.7 Mb and KASP_19405 at 677.3 Mb on 1BL (Figure 6C). The physical interval of *QYr.sdau-1BL* was reduced from 29.1 Mb to 21.6 Mb.

### 2.6. DEGs within the QYr.sdau-1BL Region

Within the region of *QYr.sdau-1BL*, 445 high-confidence genes were annotated according to the IWGSC RefSeq v1.1. Among them, 54 genes were deferentially expressed between the two parents, with 37 downregulated and 17 upregulated genes (Appendix A). Among them, 13 genes were involved in the plant–pathogen interaction pathways or encoding disease-resistance-related proteins, with seven of them upregulated in the resistant parent TTM and six downregulated (Figure 6D). Functional annotation showed that eight of the 13 genes, *TraesCS1B01G438800*, *TraesCS1B01G439400*, *TraesCS1B01G447400*, *TraesCS1B01G452600*, *TraesCS1B01G460100*, *TraesCS1B01G464400*, *TraesCS1B01G466400*, and *TraesCS1B01G466900*, encode disease-resistance proteins; three (*TraesCS1B01G448700*, *TraesCS1B01G451600*, and *TraesCS1B01G454600*) encode receptor-like protein kinases; and two (*TraesCS1B01G440700* and *TraesCS1B01G456700*) encode a WRKY transcription factor and a calcium-dependent protein kinase, respectively. Thus, these genes can be putative candidates of *QYr.sdau-1BL* (Table 3).

## 3. Discussion

During the process of wheat domestication and modern breeding, wheat landraces conserved much more genetic diversity than modern wheat varieties [19], thus providing a large source of genetic variability for identification of wheat tolerance or resistance genes to biotic and abiotic stresses. Many YR resistance genes/QTLs have been identified in wheat landraces, such as *YrHY* from the Chinese landrace “Hejiangyizai” [20], *Yr81* from the Australian wheat landrace Aus27430 [21], *YrPak* from the Pakistani landrace PI388231 [22], *QYr.GTM-5DL* from the Chinese landrace Guangtoumai [23], and *QYrhm.nwafu-2BC* from the Chinese landrace Humai 15 [24]. In the present study, the landrace TTM shows APR YR, and two QTLs, *QYr.sdau-1BL* and *QYr.sdau-6BL*, were mapped in TTM (Table 2), further indicating wheat landraces are important genetic resources for the discovery of wheat YR resistance genes or QTLs to improve YR resistance in breeding.

Although hundreds of QTLs for YR resistance have been reported [25], identifying the APR QTLs or genes are of importance for wheat production as the APR genes are more durable than race-specific genes, and especially in the Yellow and Huai River Valley Wheat Zone and Northern Winter Wheat Zone in China, the YR epidemics mainly happen at the adult-plant stage. In this study, we identified three QTLs for APR to YR on the three chromosomes: 1BL, 6BL, and 5BL.

*QYr.sdau-1BL* was mapped on 1BL in this study. Previously, multiple QTLs for APR to YR have been reported in the same position, such as *QYr.ucw-1BL* [26], *QYr.hebau-1BL.1* [17], and *QYr.cim-1BL.1* [27]. These QTLs are most likely *Yr29* for APR [28]. In a more recent study, *QYr.ucw-1BL* was further mapped to a 332-kb region between 669,902 kb and 670,234 kb [29], but the causal gene has not been identified. 

*QYr.sdau-6B* was identified on chromosome 6B in this study. Using the integrated genetic map of chromosome arm 6BL, six YR APR genes or QTLs were mapped at the similar position, including *QYr.pav-6BL* [30], *QYr.inra-6BL* [31], *YrLM168a* [32], *QYrdr.wgp-6BL.2* [33], *QYr.ucw* [34], and *QYr.nwafu-6BL* [35]. Among those reported QTLs, *YrLM168a*, *QYr.pav-6BL*, *QYr.inra-6BL*, and *QYrdr.wgp-6BL.2* overlapped with the *QYr.sdau-6B* between 500.6 Mb and 598.5 Mb (Table 2), thus they are most likely the same QTL for YR resistance.

*QYr.sdau-5B* was mapped to the region between 617.7 and 657.4 Mb on 5BL in this study (Table 2). Several studies previously also mapped QTLs for APR to YR on 5BL. For instance, *QYr.nwafu-5BL* was mapped at 670–672 Mb [36], *QYr.saas-5BL* was mapped to 564–572 Mb [37], *QYr.AYH-5BL* was mapped to 521.71–539.02 Mb [38], and *QYr.YBZR-5BL* was mapped to 519.0–542.7 Mb [39]. These QTLs were mapped in the genomic regions different from *QYr.sdau-5B*; therefore, *QYr.sdau-5B* is more likely a novel QTL for APR to YR. Interestingly, the favorable allele of *QYr.sdau-5B* is from the susceptible parent SY936, suggesting a sensitive parent can also contribute a favorable allele to YR resistance.

The advancement of next-generation sequencing greatly facilitates the high-throughput SNP genotyping and marker development in wheat. In this study, after QTL mapping using low-density GBS-derived markers, we further conducted RNA-seq to detect polymorphic SNPs to finely map the target QTL and to identify the candidate genes by analyzing the DEGs. The region of *QYr.sdau-1BL* was reduced from 29.1 Mb to 21.6 Mb by adding five RNA-seq-derived KASP markers. Moreover, RNA-seq identified 13 disease-resistance-related genes in the *QYr.sdau-1BL* region that were differently expressed between the parents TTM and SY936. Interestingly, one differentially expressed disease-resistance gene (*TraesCS1B01G454600*) was mapped within the 332-kb candidate region of *Yr29* reported by Cobo et al. [29]. This gene encodes a putative receptor-like protein kinase (RLK) and was downregulated in TTM (Table 3 and Figure 6D). RLKs are important proteins related to disease resistance, involving pathogen-associated molecular pattern (PAMP)-triggered immunity in plants [40]. Hence, these DEGs provided additional information for the identification of candidate gene of *QYr.sdau-1BL*. Our result demonstrated that QTL mapping combined with RNA-seq is effective for marker development and identification of the candidate genes underlying a QTL.

All the three QTLs identified in this study showed partial YR resistance. When the lines carried at least two of the three QTLs (*QYr.sdau-1B*, *QYr.sdau-5B*, and *QYr.sdau-6B*), the lines exhibited higher resistance to YR than the single gene lines did (Figure 4), indicating these QTLs are additive, and pyramiding of these QTLs in a single line can obtain a high level of APR. A previous study also found that epistatic interaction between *Yr29* and *QYrCW357-2AL* facilitated to enhance resistance to stripe rust [41]; thus, combining multiple APR QTLs was necessary for improvement of YR resistance in wheat MAS breeding. The markers linked to the QTLs also are useful for selecting YR resistance at the adult-plant stage in breeding. KASP has been widely used in routine MAS breeding in many breeding programs worldwide [42]. In this study, we found five KASP assays for *QYr.sdau-1BL*, which can be used to further finely map *QYr.sdau-1BL* for map-based cloning of the underlying gene, and in marker-assisted selection of *QYr.sdau-1BL* to improve YR resistance.

The resistant parent TTM carries the resistant alleles at *QYr.sdau-1B* and *QYr.sdau-6B*, but is a landrace processing many adverse agronomic traits, such as a tall plant, small kernel size, and low yield potential, which limit its direct application in breeding. The parent SY936 is a modern cultivar with a semi-dwarf stature and improved grain yield traits; the novel QTL in SY936 may be easier to use directly in breeding. However, the two QTLs in TTM can be further transferred into locally adapted cultivars to develop improved parents with these resistance QTLs for further breeding. The resistant RILs carrying two or three of the identified QTLs with improved agronomic traits—similar to or better than SY936—also can be the valuable QTL donors to be directly used as parents in breeding to improve YR resistance.

## 4. Materials and Methods

### 4.1. Plant Materials

The plant materials used for mapping comprised of 155 RILs from the cross between TTM and SY936 were developed by single-seed descent [43]. TTM is a Chinese landrace that is sensitive to *Pst* races at the seedling stage but highly resistant at the adult-plant stage, expressing the typical APR to YR; in turn, SY936 is an adapted high-yield cultivar that is highly susceptible to *Pst* races at the seedling and adult-plant stages. The sensitive cultivar Huixianhong (HXH) was used as susceptible control.

### 4.2. Seedling Stage YR Resistance Evaluation

The YR resistance of the RILs, TTM, SY936, and HXH were evaluated at the seedling stage. Around 15–20 seeds per line were sown in 50-well plastic trays (5 cm × 5 cm ×5 cm in dimension). Seedlings at the two-leaf stage were inoculated by injecting the mixture of five Chinese *Pst* races (CYR29, CYR31, CYR32, Su11, and Su14), which were obtained from the Institute of Plant Protection, Chinese Academy of Agricultural Sciences, Beijing, China. After inoculation, the seedlings were placed in a dew chamber at 10 °C with 100% relative humidity for 24 h and then moved to a temperature-controlled microclimate growth room with a daily cycle of 16 h of light at 15 °C and 8 h of darkness at 10 °C. Seedling YR responses were recorded at 18–22 days after inoculation based on a 0–9 scale [44].

### 4.3. Evaluation of Adult-Stage YR Resistance

To evaluate YR resistance at the adult-plant stage, the RILs, TTM, SY936, and HXH were planted in four consecutive seasons in the YR nursery at the Experimental Station of Shandong Agricultural University, Tai’an, China, from 2015 to 2019. All experiments were arranged in a randomized complete block design (RCBD) with two replicates. The plant materials were seeded in 2-m-long one-row plots with 5 cm between plants and 25 cm between rows. HXH was planted on every 20 rows as the susceptible control and we spread the rows to aid the fungal spread within the trial. The standard local cultivation practices were followed. Plants were inoculated every spring at the joining stage (Zadoks stage of Z33) [45] by injecting a mixture of the five prevalent Chinese *Pst* races, including *CYR29*, *CYR31*, *CYR32*, *Su11*, and *Su14*. The YR resistance of each RIL was scored for infection types (ITs) and disease severity (DS) at the adult-plant stage (Z70 stage). IT was scored using a 0–4 scale, where 0 = no visible infection or chlorosis/necrosis; 0′ = no visible uredinia with minute chlorotic/necrotic; 1 = minute or trace of restricted uredinia; 2 = small uredinia with slight sporulation; 3 = medium uredinia with moderate sporulation; and 4 = large and severe uredinia with high sporulation [46]. IT and DS were scored in 5-day intervals, from rust pustule eruption to the maximum disease severity (MDS) of YR reached on HXH (around 25 May). The area under the disease progress curve (AUDPC) was computed according to the following formula [47]: $$\mathrm{AUDPC}=\overset{n}{\underset{i}{\sum }}(X_{i}+X_{i+1})(T_{i+1}-T_{i})/2$$where *n* refers to the total number of investigations, *i* represents the *i* survey, *T_i_*_+1_ indicates the *i* + 1 survey, *T_i_* indicates the *i* survey, *X_i_*_+1_ indicates the severity of the plants surveyed in the *i* + 1 survey, and *X_i_* indicates the severity of the plants surveyed in the *i* survey.

### 4.4. Statistical Analysis 

Pearson’s correlation coefficients (*r*) were calculated for IT, MDS, and AUDPC across locations and years using SPSS version 20 (http://www.spss.com (accessed on 5 Dec. 2021)). ANOVA of the phenotypic data and heritability were analyzed using the “aov” module implemented in QTL IciMapping V4.1 software [48]. 

### 4.5. Linkage Map Construction and QTL Mapping

Genotyping of the RIL population by GBS and SSR markers has been described previously [43,49]. A linkage map with combined SNP and SSR markers was constructed by Lin et al. [43] and used to map the QTLs for YR resistance in this study. The composite interval mapping function in WinQTLCart v2.5 was used for QTL analysis [50]. Significant QTLs were claimed if the LOD scores were above the thresholds determined by 1000 permutations. 

### 4.6. RNA-Seq

When TTM and SY936 showed significant differences in YR resistance in the 2016–2017 field experiment, their leaves were collected independently to extract the total RNA using the Illumina TruSeq RNA Sample Prep Kit (Illumina, Inc., San Diego, CA, USA). RNA-Seq samples were sequenced using the Illumina HiSeq 4000 platform (Beijing Southern Genome Research Technology Co., Ltd., Beijing, China). The raw reads generated were filtered by Trimmomatic v0.36 software [51]. The clean RNA-Seq reads were aligned to the Chinese Spring reference genome RefSeq v1.1 [52] (http://www.wheatgenome.org/ (accessed on 20 Dec. 2018)) using Hisat [53]. SNPs and Indels were called using GATK [54]. The differently expressed genes (DEGs) were identified by DEGseq [55] based on FPKM analyzed by HTseq [56]. The genes with a difference fold ≥ 2.0 and Q-value ≤ 0.01 were considered as DEGs between parents. The analysis of the enriched gene ontology terms (GO) was conducted using GOseq [57].

### 4.7. KASP Assays 

Resistance-related polymorphic SNPs were converted into KASP markers using Poly Marker (http://www.polymarker.info/ (accessed on 10 Sep. 2019)), and were screened in the RIL population. Protocols for preparation of the KASP reactions followed the KASP manual (https://biosearch-cdn.azureedge.net/assetsv6/Analysis-of-KASP-genotyping-data-using-cluster-plots.pdf (accessed on 10 Nov. 2019)). Assays were carried out in 384-well plates with 6 μL PCR reaction volumes consisting of 3 μL of a 2× KASP master mix, 0.0825 μL of a KASP primer assay mix, and 3 μL of genomic DNA at a concentration of 20 ng/μL. The PCR was conducted with a hot start at 94 °C for 15 min, followed by 10 touchdown cycles (94 °C for 20 s; touchdown at 65 °C initially and decreasing by 0.8 °C per cycle for 25 s), followed by 30 additional cycles of denaturation (94 °C for 10 s) and annealing/extension (57 °C for 60 s). The KASP end-point fluorescent images were visualized using the FLUOstar Omega microplate reader (BMG Labtech, Durham, NC, USA).

## Figures and Tables

**Figure 1 ijms-23-09662-f001:**
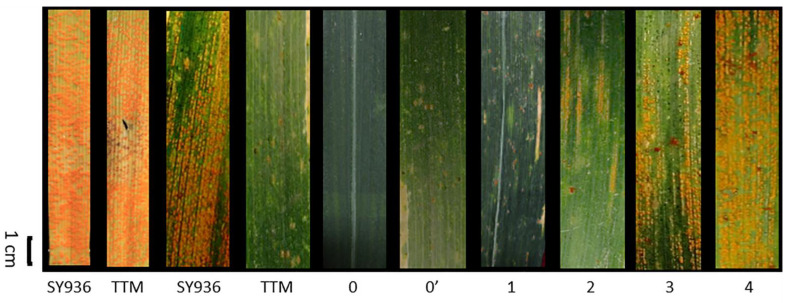
Infection types (ITs) of stripe rust resistance for parents and recombinant inbred lines (RILs). From left are yellow rust (YR)-inoculated leaves, showing YR symptoms on the parents SY936 and TTM at the seedling (leaves 1–2) and adult-plant stages (leaves 3–4), and the recombinant inbred lines, with an IT of 0 (immune), 0′ (near immune), 1 (highly resistant), 2 (moderately resistant), 3 (moderately susceptible), and 4 (highly susceptible) at the adult stage.

**Figure 2 ijms-23-09662-f002:**
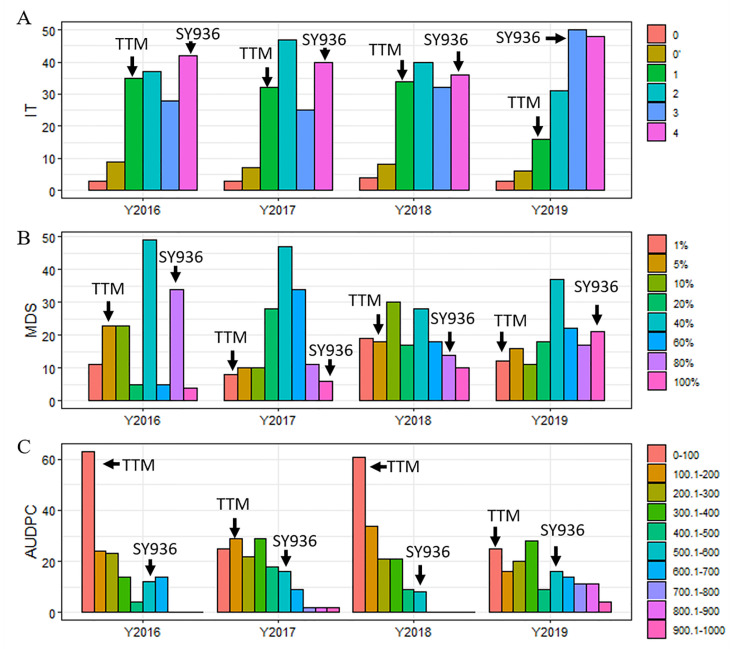
Distribution of infection types (**A**), maximum disease severity (**B**), and area under the disease progress curve (**C**) for the RIL population evaluated in four field experiments from 2016–2019. TTM and SY936 refer to the stripe-rust-tolerant parent Tutoumai and susceptible parent Siyang 936, respectively.

**Figure 3 ijms-23-09662-f003:**
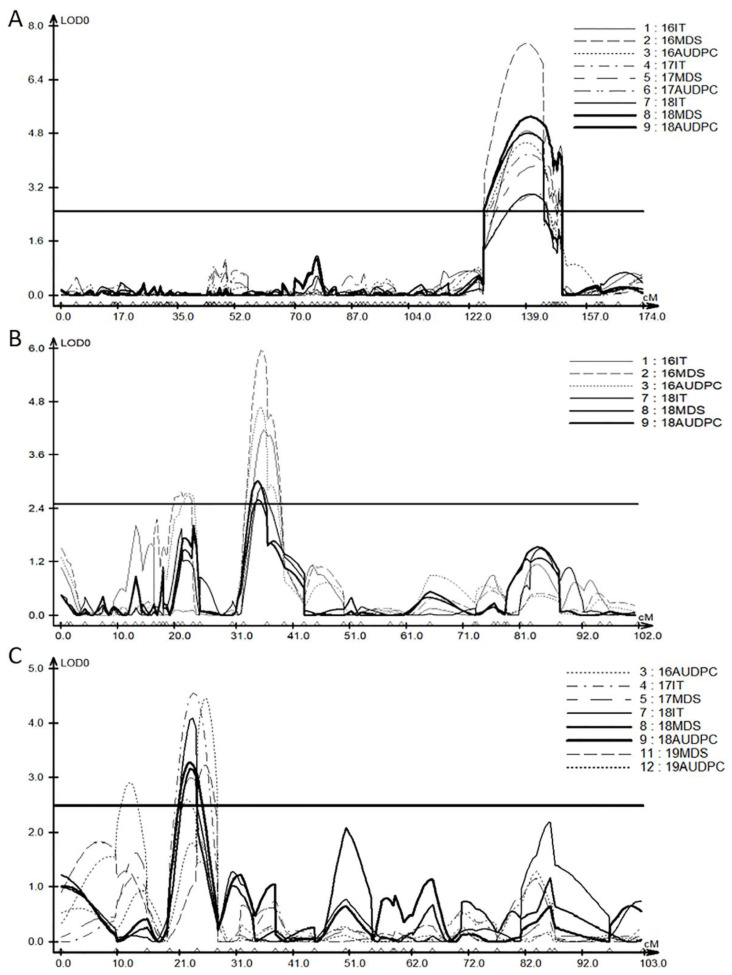
Composite interval mapping (CIM) of QTLs for stripe rust adult-plant resistance on chromosomes 1B (**A**), 5B (**B**), and 6B (**C**), using phenotypic data collected from the four experiments from 2016 to 2019. The line parallel to the *X*-axis is the threshold line for the significant LOD score of 2.5. Genetic distances are shown in centiMorgans (cM). IT = infection type; MDS = maximum disease severity; AUDPC = area under the disease progress curve.

**Figure 4 ijms-23-09662-f004:**
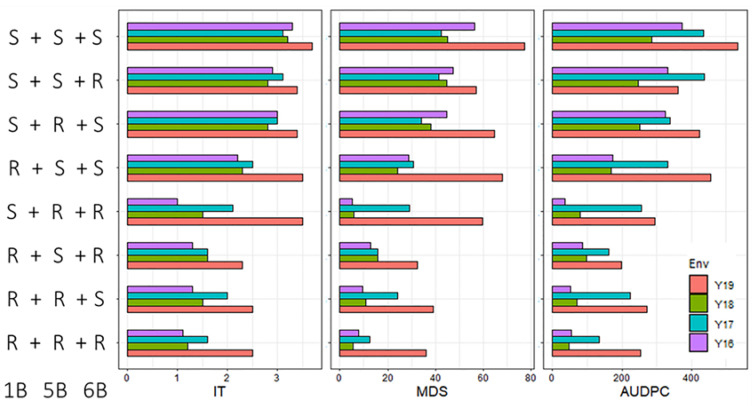
The accumulative effects of the QTLs identified in this study. 1B, 5B, and 6B indicate the three QTLs *QYr.sdau-1B*, *QYr.sdau-5B*, and *QYr.sdau-6B*, respectively. R and S indicate the resistance and susceptibility alleles of the identified QTLs. IT = infection type; MDS = maximum disease severity; AUDPC = area under the disease progress curve.

**Figure 5 ijms-23-09662-f005:**
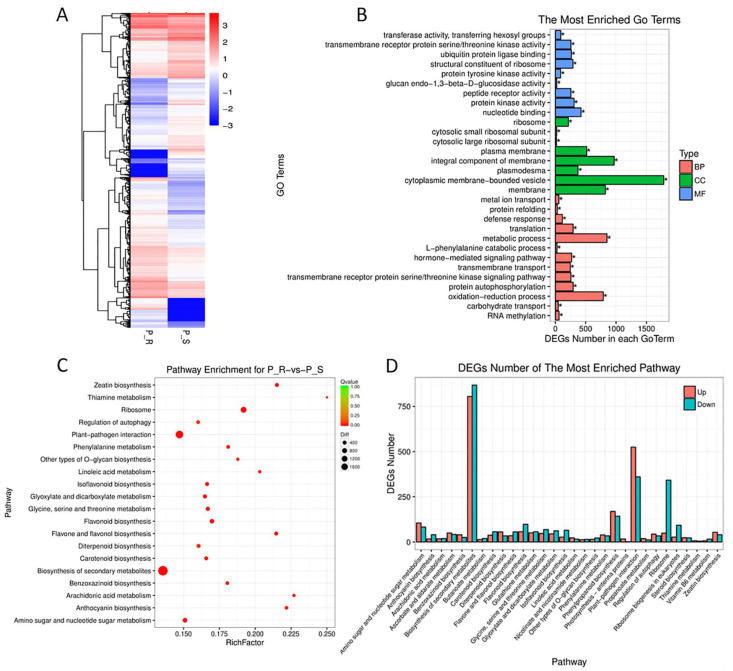
Gene ontology (GO) analysis of the differentially expressed genes (DEGs) between Tutoumai (TTM) and Siyang 936 (SY936). (**A**) Cluster heat map of DEGs between the resistant parent TTM (P_R) and susceptible parent SY936 (R_S). (**B**) The most enriched GO terms of the DEGs. (**C**) Pathway enrichment of the DEGs. (**D**) Number of DEGs of the most enriched pathway.

**Figure 6 ijms-23-09662-f006:**
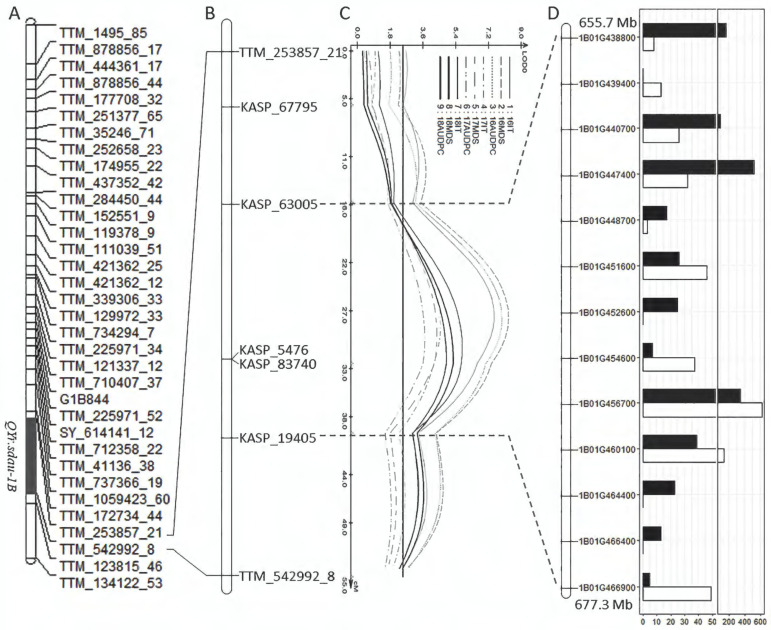
A physical map of *QYr.sdau-1B*. (**A**) The position of *QYr.sdau-1B* in a genetic mapping. (**B**) Five KASP markers developed based on the SNPs in the *QYr.sdau-1B* region identified from RNA-seq. (**C**) Re-mapped location of *QYr.sdau-1B* after incorporation of the new KASP markers. (**D**) The expression profiles of 13 disease-resistance-related genes identified in the region of *QYr.sdau-1B* from RNA-seq. The white and black bars indicated the relative expression levels of alleles from the susceptible parent Siyang 936 and resistant parent Tutoumai, respectively; the *x*-axis shows the FPKM value of the genes in the region.

**Table 1 ijms-23-09662-t001:** Analysis of variance of the three stripe-rust-resistance traits evaluated at the adult-stage under four environments.

Trait	Factor	Df	SS	MS	F-Value	*p*-Value	Heritability
IT	Genotype	145	947.0	6.5	22.6	2.9 × 10^−138^	0.83
	Environment	3	136.3	45.4	157.5	4.3 × 10^−69^	
	G × E interaction	435	365.3	0.8	2.9	6.8 × 10^−28^	
	Replication/Env	3	3.8	1.3	4.4	0.0044	
	Residuals	435	125.5	0.3			
MDS	Genotype	145	42,4371.7	2926.7	28.2	3.9 × 10^−156^	0.80
	Environment	3	102,679.6	34,226.5	329.9	1.2 × 10^−111^	
	G × E interaction	435	195,868.6	450.3	4.3	4.0 × 10^−149^	
	Replication/Env	3	3215.9	1072.0	10.3	1.4 × 10^−6^	
	Residuals	435	45,129.7	103.7			
AUDPC	Genotype	145	25,294,308.9	174,443.5	36.2	4.8 × 10^−177^	0.83
	Environment	3	4,829,528.6	1,609,842.9	334.3	1.7 × 10^−112^	
	G × E interaction	435	9,435,447.8	21,690.7	4.5	2.5 × 10^−51^	
	Replication/Env	3	347,671.7	115,890.6	24.1	2.0 × 10^−14^	
	Residuals	435	2,095,044.3	4816.2			

**Table 2 ijms-23-09662-t002:** Name, position, LOD score, phenotypic variation explained (PVE), and additive effects of quantitative trait loci (QTLs) for stripe-rust resistance at the adult stage in a population of Tutoumai x Siyang 936 identified using composite interval mapping (CIM) and the phenotypic data collected from the experiments conducted from 2016 to 2019.

QTL	Genetic Distance (cM)	Physical Distance (Mb)	Year-Trait	LOD	PVE (%)	Add
*QYr.sdau-1B*	126.3–150.7	653.0–682.1	16IT	4.3	8.0	−1.1
			16MDS	7.5	17.7	−21.3
			16AUDPC	6.8	21.2	−97.5
			17IT	4.2	9.7	−0.5
			17MDS	4.2	11.0	−10.8
			17AUDPC	2.7	13.5	−76.9
			18IT	2.8	13.8	−0.4
			18MDS	2.6	10.8	−8.9
			18AUDPC	2.8	9.2	−46.5
*QYr.sdau-5B*	32.2–39.4	617.7–657.4	16IT	4	14.7	0.7
			16MDS	5.9	22.7	19.5
			16AUDPC	4.6	19.2	129.3
			18IT	2.6	10.1	0.5
			18MDS	2.6	12.1	12.4
			18AUDPC	3.0	13.4	74.1
*QYr.sdau-6B*	19.2–27.6	500.6–598.5	16AUDPC	2.6	12.1	−87.5
			17IT	4.5	16.5	−0.5
			17MDS	2.9	12.4	−8.9
			18IT	4.5	18.0	−0.7
			18MDS	3.2	13.7	−14.3
			18AUDPC	3.0	11.6	−69.4
			19MDS	3.2	12.1	−14.7
			19AUDPC	4.4	16.4	−131.5

**Table 3 ijms-23-09662-t003:** The 13 differently expressed genes (DEGs) in the region of *QYr.sdau-1B* that relate to disease resistance.

GeneID	Functional Annotation
*TraesCS1B01G438800*	Disease-resistance protein (TIR-NBS-LRR class) family
*TraesCS1B01G439400*	Disease-resistance protein (TIR-NBS-LRR class) family
*TraesCS1B01G440700*	WRKY transcription factor
*TraesCS1B01G447400*	Disease-resistance protein RPM1
*TraesCS1B01G448700*	Receptor-like protein kinase
*TraesCS1B01G451600*	Receptor-like protein kinase
*TraesCS1B01G452600*	Disease-resistance protein (CC-NBS-LRR class) family
*TraesCS1B01G454600*	Receptor-like protein kinase, putative, expressed
*TraesCS1B01G456700*	Calcium-dependent protein kinase
*TraesCS1B01G460100*	Disease-resistance protein (NBS-LRR class) family
*TraesCS1B01G464400*	Disease-resistance protein RGA2
*TraesCS1B01G466400*	Disease-resistance protein
*TraesCS1B01G466900*	Disease-resistance protein

## Data Availability

The data presented in this study are available on request from the corresponding author.

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
