# Peer review of "Mapping QTL for Adult-Plant Resistance to Stripe Rust in a Chinese Wheat Landrace"

_ijms, 2022, doi:10.3390/ijms23179662_

Round 1
Reviewer 1 Report
My comments can be found in the attached MS.

Reviewer 2 Report
The manuscript (Mapping QTL for Adult Plant Resistance to Stripe Rust in Chinese Wheat Landrace) is interesting and introduces valuable information for wheat breeding programs. I wonder if the experiments were done during the seasons 2015- 2019, but the manuscript was submitted after three years, and everything was changed; the environmental conditions and the resistance and susceptible lines to Pst could be changed due to the appearance of new races of the pathogen. Could you please explain? do you think that your study results are still dependent for researchers?
Several details are missing in the materials and methods, for example:
The concentration of spores before artificial inoculation
Details about disease scale
Did you use a pathogen spreader line from the susceptible cultivar in the field experiments?
L34 Please update the sentence to (Large economic losses caused by YR epidemics usually occur in the major wheat production countries).
Please correct the following:
L35 Please change to (the major wheat producing countries).
L36 Please change to (poses a serious threat).
L47 Please change (to most of the cataloged Yr genes)
L49 Please change to (have a short duration of effectiveness)
L53 Please change to high-density SNP chips have revolutionized.
L73 Please change to (TTM and SY936 were developed by single-seed descent).
L74 Please change to (Pst races at the seedling).
L88 Please change to (To evaluate YR resistance at the adult plant stage).
L95 Please change to (at the joining stage).
L120 Please change to (SY936 showed significant differences in YR resistance).
L233 Please correct phenylpropanoid biosynthesis
The discussion looks like the results; I suppose to read a deep discussion of your study results. Please improve the discussion keeping and following the Journal format.
Reviewer 3 Report
The manuscript reported the novel QTL loci for adult plant resistance to stripe rust in wheat lines, and new KASP markers were identified for breeding through MAS. The result is interesting for wheat geneticists and breeders of disease resistance. The study was clearly presented and the conclusions were supported by multiple evidences from comprehensive QTL mapping of resistance survey and GBS analysis.
This manuscript should be suitable for publication after replying below questions and considering revision:
(1) Whether the parent cultivar SY936 has 1RS.1BL translocation or not may be proved.
(2) The data for RNAseq should be deposited in a public database and the accession or project number should be included in the manuscript.
(3) The recent stripe rust QTL studies including Yr29 regions may be suggested to be cited.
(4) A publication of "Rosewarne GN, Herrera‑Foessel SA, Singh RP, Huerta‑Espino J, Lan CX, He ZH. Quantitative trait loci of stripe rust resistance in wheat. Theor Appl Genet (2013) 126:2427–2449" will be suggested to be cited in Discussion part.
Round 2
Reviewer 2 Report
Thank you very much. The corrections were done.